# Daily Administered Dual-Light Photodynamic Therapy Provides a Sustained Antibacterial Effect on *Staphylococcus aureus*

**DOI:** 10.3390/antibiotics10101240

**Published:** 2021-10-13

**Authors:** Sakari Nikinmaa, Anna Podonyi, Peter Raivio, Jukka Meurman, Timo Sorsa, Juha Rantala, Esko Kankuri, Tuomas Tauriainen, Tommi Pätilä

**Affiliations:** 1Department of Neuroscience and Biomedical Engineering, Aalto University, 02150 Espoo, Finland; sakari@koitehealth.com; 2Koite Health Oy, 02150 Espoo, Finland; juha.rantala@koitehealth.com; 3Department of Cardiac Surgery, University Hospital Southampton, Southampton SO16 6YD, Hampshire, UK; anna.podonyi@gmail.com; 4Heart and Lung Center, Meilahti Hospital, 00290 Helsinki, Finland; peter.raivio@hus.fi (P.R.); tuomas.ttau@gmail.com (T.T.); 5Department of Oral and Maxillofacial Diseases, University of Helsinki, 00290 Helsinki, Finland; jukka.meurman@helsinki.fi (J.M.); timo.sorsa@helsinki.fi (T.S.); 6Department of Pharmacology, University of Helsinki, 00290 Helsinki, Finland; esko.kankuri@helsinki.fi; 7Department of Congenital Heart Surgery and Organ Transplantation, New Children’s Hospital, University of Helsinki, 00290 Helsinki, Finland

**Keywords:** biofilm, *Staphylococcus aureus*, antibacterial photodynamic therapy

## Abstract

New means to reduce excessive antibiotic use are urgently needed. This study tested dual-light aPDT against *Staphylococcus aureus* biofilm with different relative ratios of light energy with indocyanine green. We applied single-light aPDT (810 nm aPDT, 405 aBL) or dual-light aPDT (simultaneous 810 nm aPDT and 405 nm aBL), in both cases, together with the ICG photosensitizer with constant energy of 100 or 200 J/cm^2^. Single-dose light exposures were given after one-day, three-day, or six-day biofilm incubations. A repeated daily dose of identical light energy was applied during biofilm incubations for the three- and six-day biofilms. Using 100 J/cm^2^ light energy against the one-day biofilm, the dual-light aPDT consisting of more than half of aBL was the most effective. On a three-day maturated biofilm, single-dose exposure to aPDT or dual-light aPDT was more effective than aBL alone. With total light energy of 200 J/cm^2^, all dual-light treatments were effective. Dual-light aPDT improves the bactericidal effect on *Staphylococcus aureus* biofilm compared to aPDT or aBL and provides a sustained effect. An increase in the relative ratio of aBL strengthens the antibacterial effect, mainly when the treatment is repeatedly applied. Thus, the light components’ energy ratio is essential with dual-light.

## 1. Introduction

*Staphylococcus* is a genus of Gram-positive bacteria in the family *Staphylococcaceae* in the order *Bacillales*. Despite being part of the harmless skin microbial flora, *Staphylococcus aureus* is potentially the most dangerous *staphylococcal* bacteria. Skin infections by *S. aureus* are common and generally benign, but potentially lethal infections are due once the bacteria enter the bloodstream. [1,2]. Indeed, *S. aureus* is the leading cause of infectious endocarditis [3,4]. The morbidity and mortality from *S. aureus* bacteremia have only minor improvement when comparing the reports from the 1940s to 2010s [5,6]. Roughly twenty percent of the general population [7] and 60% of healthcare professionals carry [8] the bacteria. 

Local *S. aureus* infections are commonly treated with peroral antibiotics. However, the use of antibiotics leads to antimicrobial resistance formation, which is one of the most critical threats to modern medicine. *S. aureus* is notorious for its ability to generate resistance, and the resistant strains are widely spread worldwide [9,10,11]. Multiple challenges relate to the novel antibiotic drug discovery and development [12,13], and diminishing investments for research and development of new antibiotics are reported [14]. The need for new solutions has never been more urgent. 

Antimicrobial photodynamic therapy (aPDT) has arisen as an alternative method against microbial infections [15,16,17]. The antimicrobial effect of aPDT is based mainly on the principle that visible light activates an externally applied photosensitizer, resulting in the generation of reactive oxygen species (ROS) that kill bacteria unselectively via an oxidative burst [17]. Antimicrobial blue light (aBL) is suggested to be based on the same principle. However, the photosensitizers in the latter process are inherent molecules within the bacteria itself, such as porphyrins and flavins. [18]. During the last decades, the development of light technology has provided an opportunity for extensive in vitro research, which is now ready to be translated into clinical practice. There is a notable scarcity of papers discussing a combination of antibacterial photodynamic therapy and antibacterial blue light in the literature. Indeed, we have recently published the efficacy of combined aPDT and aBL against streptococcus biofilm, with indocyanine green (ICG) as the photosensitizer. The inherent lack of catalase enzyme in streptococci, which is the bacteria’s main factor in clearing hydrogen peroxide-type reactive oxygen species from the cell, makes *streptococcus* very vulnerable to aPDT. However, simultaneous application of aBL increased the bactericidal effect significantly [19]. These results encourage the assessment of the treatment method on other clinically important pathogens, as previously discussed, such as *Staphylococcus aureus.* To the best of our knowledge, our study group is the first to describe the antibacterial effects of dual-light antibacterial therapy combining 405 nm aBL with 810 nm aPDT.

In the present study, we investigated, for the first time, the use of dual-light aPDT against *S. aureus*. Most papers regarding aPDT are reporting antibacterial effects of a single wavelength. The improvent in efficacy by using dual-light against streptococci might also be valid against staphylococci. Thus, dual-light aPDT could be used instead of antibiotics for superficial skin infections that are often caused by streptococci and staphylococci. Furthermore, this technique might be an accessory tool in the treatment of more severe conditions, such as infective endocarditis, which requires surgical debridement via sternotomy.

## 2. Materials and Methods

The present study aimed to investigate the effects of aBL (405 nm), aPDT (810 nm combined with a photosensitizer: ICG), and dual-light antibacterial photodynamic therapy (405 and 810 nm combined with a P.S.) on monospecies *S. aureus* ATCC 25923 incubated for one day before any treatment. This strain is a known biofilm-forming strain [20]. The formed biofilms were divided into subgroups according to the total incubation time and the treatment method. The experiments and the number of assays are shown in Table 1.

Apart from the species of bacteria studied, the study protocol, details on the biofilm models, and light exposures, in addition to information on colony-forming-unit counting used in the present study, have also been presented accurately elsewhere [19,21]. 

The methodology in the present study is presented in the following steps. Step 1: The *Staphylococcus aureus* strain (ATCC 25923) growth in a BHI broth (Bio-Rad 3564014, Bio-Rad Laboratories Inc., Hercules, CA, USA). Step 2: Incubation at +36 degrees C in an air concentration of 5% CO^2^ for 18 h (NuAire DH autoflow 5500, NuAire Inc., Minneapolis, MN, USA). Step 3: Dilution to the optical density of 0.46, using 0.9% NaCl solution (measured by using a spectrophotometer, Varian Cary 100 Bio UV–VIS, Agilent Technologies, Inc., Santa Clara, CA, USA, and Den 1 McFarland Densitometer, Biosan, Riga, Latvia). Step 4: Biofilm growth in a flat-bottom 96-well plate (Thermo Fisher Scientific Inc., Waltham, MA, USA). Step 5: Placing 100 μL of *S. aureus* suspension into the well-plates, each primed with 100 μL of BHI broth. (Additional step, Step 5.1: Incubation at a temperature of 36 degrees C and in an air concentration of 5% CO^2^, in addition to a daily change of the 100 μL BHI broth to a fresh solution.) Step 6: Removal of Growth medium and replacement with indocyanine green solution 250 μg/mL (Verdye, Diagnostic Green, GmBH, Aschheim, Germany). Step 7: Incubation in a dark room for 10 min. Step 8: Washing the biofilm with 0.9% NaCl solution until each well contained a total volume of 200 μL. Step 9: Light exposure, using a custom-made LED light (Lumichip Oy, Espoo, Finland); the time of exposure was calculated by using the known irradiances of the light sources (measured by using a light energy meter, Thorlabs PM 100D and an S121C sensor head, Thorlabs Inc., Newton, NJ, USA, in addition to a spectroradiometer, BTS256, Gigahertz-Optik GmBH, Türkenfeld, Germany). The irradiances of used light, the wavelengths, the number of treatments, and incubation repeats in each biofilm study are presented in Table 1. For example, irradiance times of 1250 and 1000 s were required to reach the desired radiant exposure of 100 J/cm^2^ for the aBL and aPDT, respectively. Step 10: Changing of the BHI broth and subsequent incubation or biofilm removal for colony-forming-unit counting if the planned study exposure was finished. Step 11: After the light exposure, removal of the entire biofilms into a 1 mL test tube, forming 200 μL of solution. Step 12: Vortexing by using a (Vortex-Genie, Scientific Industries Inc., Bohemia, NY, USA) and serial dilution ranging from 1:1 to 1:100,000, using sterile tipped ART filters (Thermo Fisher Scientific Inc., Waltham, MA, USA). Step 13: Spreading of 100 μL of the resulting biofilm dilution onto a BHI agar plate, using a sterile L-rod. The dilutions were performed according to the observed biofilm mass from the well-plates. Dilution, where the CFU counts were between 30 and 800, were selected for analysis, whereas the results for the CFU 0 analysis were obtained from a solution with a 1:1 dilution factor. Step 14: Incubation for 48 h and photographing of the plates (Leica TCS CARS SP 8X microscope, Leica Microsystems, Wetzlar, Germany) with HC PL APO CS2 20X/0.75 numerical-aperture multi-immersion and HX PL APO CS2 63X/1.2 numerical-aperture water-immersion objectives. Step 15: Staining with a live/dead BacLight bacterial viability kit (Molecular Probes, Invitrogen, Eugene, OR, USA) and incubation in a dark room for 15 min. Step 16: Examination under a confocal scanning laser microscope, with a two-laser system (488 nm argon laser and a 561 nm DPSS laser). The emission windows for the 488 and 561 nm lasers were set at 500–530 nm and 620–640 nm, respectively.

### 2.1. Outcome Endpoints

The primary outcome endpoints in the present study are the number of viable colony-forming units observed after treatment with aBL, aPDT, or dual-light aPDT and the differences in efficacy between the treatment methods.

### 2.2. Statistical Analysis

Statistical analysis was performed by using GraphPad Prism 8 software (GraphPad Software, San Diego, CA, USA). The continuous variable of colony-forming units was reported as medians. Mann–Whitney U test was used for all univariate analyses. The tests were two-tailed, and a *p*-value < 0.05 represented statistical significance.

## 3. Results

### 3.1. One-Day S. aureus Biofilm Treated Once by 100 J/cm^2^

The one-day *S. aureus* biofilm showed a significant reduction in viability when treated with aBL. A decrease from the median of 1.3 × 10^9^ CFUs (the range being 1.0 × 10^9^–3.3 × 10^9^ CFUs, with three assays) in the control biofilm to a median of 3.1 × 10^6^ CFUs (ranging between 8.0 × 10^5^ and 1.0 × 10^7^ CFUs, with six assays), with *p* = 0.024. An aPDT treatment with an 810 nm light combined with ICG photosensitizer resulted in the efficacy of a similar scale when compared to aBL. The median number of CFUs was 2.7 × 10^5^ CFUs (the range being 6.0 × 10^4^–3.4 × 10^6^ CFUs, with six assays), with *p* = 0.024. However, the dual-light combination of one part of aBL and three parts of aPDT, at irradiances of 42 and 135 mW/cm^2^, respectively, decreased the median of alive bacteria to 3.6 × 10^4^ CFUs (the range being 1.9 × 10^3^–2.4 × 10^5^ CFUs with six assays), with *p* = 0.024. The bactericidal effect of dual-light with a 1:1 ratio of aBL and aPDT, (with irradiances of 79 and 73 mW/cm^2^, respectively) decreased the median number of living bacteria to 6.0 × 10^4^ CFUs (the range being 3.0 × 10^2^–2.6 × 10^4^ CFUs with six assays), with *p* = 0.024. When the amount of aBL was raised to three parts, and aPDT was lowered to one part in the dual-light system with respective irradiances of 130 and 38 mW/cm^2^, the bacterial viability reduced to 5.0 × 10^2^ (the range being 2.0 × 10^2^–1.6 × 10^3^ CFUs with six assays), with *p* = 0.024 (Figure 1). 

### 3.2. Three-Day S. aureus Biofilm Treated Once by 100 J/cm^2^

The three-day *S. aureus* biofilm presented a significantly reduced viability after a single application of aBL, from the control biofilm median of 3.0 × 10^11^ CFUs (the range being 2.9 × 10^11^–3.3 × 10^12^ CFUs, with three assays) to a median of 1.0 × 10^4^ CFUs (the range being 4.9 × 10^6^–3.4 × 10^8^ CFUs, with six assays), with *p* = 0.024. The use of aPDT showed a slightly lesser degree of antibacterial efficiency than aBL, with an observed median of 2.7 × 10^8^ CFUs (the range being 1.0 × 10^8^–3.8 × 10^8^ CFUs, with six assays), with *p* = 0.024. An exposure of dual-light in a combination of one part of aBL and three parts of aPDT (at irradiances of 42 and 135 mW/cm^2^, respectively) decreased the median of live bacteria to 7.1 × 10^6^ CFUs (the range being 3.1–2.3 × 10^7^ CFUs with six assays), with *p* = 0.024. When the dual-light fractions were administered in a 1:1 ratio (with irradiances of 79 and 73 mW/cm^2^, respectively), the median of live bacteria decreased to 3.2 × 10^7^ CFUs (the range being 9.4 × 10^6^–4.0 × 10^7^ CFUs with six assays), with *p* = 0.024. After increasing the amount of aBL to three parts and decreasing aPDT to one part in the dual-light system (with respective irradiances of 130 and 38 mW/cm^2^), the median number of colony-forming units was 5.0 × 10^3^ (the range being 0.0–7.0 × 10^3^ CFUs with 12 assays), with *p* = 0.024 (see Figure 2).

### 3.3. Three-Day S. aureus Biofilm Treated Daily by 100 J/cm^2^

When the three-day *S. aureus* biofilm was exposed to daily antibacterial therapy, the daily applied aBL significantly reduced the number of bacterial CFUs to a median of 7.5 × 10^7^ (ranging between 1.2 × 10^7^ and 3.5 × 10^8^ CFUs, with six assays) when compared to the control biofilm median of 2.8 × 10^11^ CFUs (the range being 1.9 × 10^11^–2.9 × 10^11^ CFUs, with three assays), with *p* = 0.024. APDT resulted in a similar antibacterial efficiency, presenting with a median of 3.0 × 10^7^ CFUs (the range being 9.5 × 10^6^–1.6 × 10^8^ CFUs, with six assays), with *p* = 0.024. The dual-light combination of one part of aBL and three parts of aPDT, at irradiances of 42 mW/cm^2^ and 135 mW/cm^2^, respectively, decreased the median of live bacteria to 4.4 × 10^6^ CFUs (a range of 9.0 × 10^5^–1.0 × 10^7^ CFUs with six assays), with *p* = 0.024. The one-to-one ratio of aBL and 810 nm PDT, with irradiances of 79 and 73 mW/cm^2^, respectively, decreased the median of live bacteria to 1.6 × 10^7^ CFUs (the range being 3.8 × 10^6^–2.6 × 10^7^ CFUs with six assays), with *p* = 0.024. When the amount of aBL was raised to three parts, and aPDT was set at one part in the dual-light system with respective irradiances of 130 and 38 mW/cm^2^, the bacterial viability was reduced to 10 CFUs (ranging between 0 and 140 CFUs with six assays), with *p* = 0.0012 (see Figure 3).

### 3.4. Six-Day S. aureus Biofilm Treated Daily by 100 J/cm^2^

Six-day *S. aureus* biofilm was exposed to daily applied antibacterial light therapy. The dual-light combination of one part of aBL and three parts of aPDT, at irradiances of 42 and 135 mW/cm^2^ respectively, decreased the bacterial viability from the control biofilm of 1.0 × 10^10^ CFUs (the range being 8.0 × 10^9^–3.2 × 10^10^, with three assays) to a median number of 7.0 × 10^5^ CFUs (ranging between 5.4 × 10^4^ and 4.4 × 10^6^ CFUs with six assays), with *p* = 0.024. Again, the one-to-one ratio of aBL and 810 nm PDT, with irradiances of 79 and 73 mW/cm^2^, respectively, decreased the median of live bacteria to 2.2 × 10^7^ CFUs (the range being 1.5 × 10^7^–3.8 × 10^7^ CFUs with six assays), with *p* = 0.024. When the amount of aBL was set at three parts, and aPDT was reduced to one part of the total amount of light, with irradiances of 130 and 38 mW/cm^2^, respectively, the number of bacterial colony-forming units reduced to a median of 0 (with a range of 0–61 CFUs with six assays), with *p* = 0.012 (see Figure 4).

### 3.5. One-Day S. aureus Biofilm Treated Once by 200 J/cm^2^

The one-day *S. aureus* biofilm showed a significant reduction in viability when treated with aBL at an energy density of 200 J/cm^2^. The median number of CFUs decreased from 9.4 × 10^8^ CFUs (with a range of 3.7 × 10^8^–3.3 × 10^9^ CFUs, with six assays) in the control test to 1.5 × 10^3^ CFUs (ranging between 8.0 × 10^2^ and 3.8 × 10^3^ CFUs, with six assays), with *p* = 0.0022. APDT resulted in a similar scale efficiency as did aBL, decreasing the median number of CFUs to 1.8 × 10^3^ (the range being 6.0 × 10^2^–2.2 × 10^3^ CFUs, with six assays), with *p* = 0.0022. However, the dual-light combination of one part of aBL and three parts of aPDT, at irradiances of 42 and 135 mW/cm^2^, respectively, decreased the median of live bacteria to 1.0 × 10^2^ CFUs (ranging between 0.0 and 2.0 × 10^2^ CFUs with six assays), with *p* = 0.0022. The bactericidal effect of dual-light with a 1:1 ratio of aBL and aPDT (with irradiances of 79 and 73 mW/cm^2^, respectively) decreased the median number of CFUs to 50 (the range being 0.0–100.0 CFUs with six assays), with *p* = 0.0022. When the amount of aBL was raised to three parts, and aPDT was set at one part in the dual-light system, with respective irradiances of 130 mW/cm^2^ and 38 mW/cm^2^, the bacterial viability was reduced to 100.0 (the range being 0–200.0 CFUs with six assays), with *p* = 0.0022 (see Figure 5).

### 3.6. Three-Day S. aureus Biofilm Treated Daily by 200 J/cm^2^

When the three-day *S. aureus* biofilm was exposed to the daily antibacterial therapy, aBL reduced the bacterial viability to almost zero. The median number of CFUs with aBL was 0 (ranging between 0.0 and 210.0 CFUs, with six assays), and the median number of CFUs in the control test was 4.2 × 10^10^ (the range being 1.6 × 10^10^–5.2 × 10^10^ CFUs, with six assays), with *p* = 0.0022. Similarly, aPDT resulted in a significant reduction in viability as did aBL, presenting with a median of 0.5 CFUs (the range being 0–3.6 × 10^3^ CFUs, with six assays), with *p* = 0.0022. The dual-light combination of one part of aBL and three parts of aPDT, at irradiances of 42 mW/cm^2^ and 135 mW/cm^2^, respectively, decreased the number of living bacteria to 0 CFUs (with a range of 0 CFUs with six assays), with *p* = 0.0022. The one-to-one ratio of aBL and 810 nm PDT, with irradiances of 79 and 73 mW/cm^2^, respectively, decreased the median number of CFUs to 0 (the range being 0–1 CFUs with six assays), with *p* = 0.0022. When the amount of aBL was raised to three parts, and aPDT was set at one part in the dual-light system, with respective irradiances of 130 and 38 mW/cm^2^, 0 CFUs were found (the range being 0 CFUs with six assays), with *p* = 0.0022 (see Figure 6).

## 4. Discussion

This study demonstrates the efficacy of dual-light aPDT against *S. aureus* biofilm when compared to separately applied 810 nm aPDT with ICG photosensitizer or 405 nm aBL. We tested three different combinations of the dual-light to find out the most effective combination against *S. aureus*. To enable direct comparison between the groups, we kept the radiant exposure constant at 100 J/cm^2^ and examined the effect of doubling the radiant exposure to 200 J/cm^2^. The combination of dual-light, where aBL was increased, showed the most effective antibacterial combination against *S. aureus*. On the one-day biofilm model, the single-dose light exposures at 100 J/cm^2^ resulted in a decrease of six logarithmic scales in the bacterial count when compared to the control biofilm. The decrease in CFU formation was three logarithmic scales more efficient when compared to aPDT alone and four logarithmic scales more efficient than aBL. Similar results were seen when the three-day biofilm was treated with a single dose of dual-light. When the light exposure was doubled up to 200 J/cm^2^, the antibacterial efficacy increased expectedly in all the treatment groups, but again, the most effective antibacterial action was seen in the dual-light-treated groups. 

We also tested protocols where the *S. aureus* biofilm was treated daily, simulating clinical treatment scenarios. Investigating repeated treatment effects in biofilm is important. Although aPDT treatment is assumed to be unable to cause resistance [22], the bacteria are capable of adapting to environmental stress [23]. For example, Guffrey et al. (2013) reported a decrease in the inactivation effectiveness of 405 nm blue light against in vitro *S. aureus* after the fifth bacterial generation was exposed to blue light [24]. However, in the present study, repeated daily treatment with 100 J/cm^2^ radiant exposure resulted in a persistent antibacterial effect, especially in settings where the relative amount of aBL was higher. When the total daily applied radiant exposure increased to 200 J/cm^2^, the antibacterial impact increased to the disinfection level in all treatment groups. 

Opposite to UV-light bacterial inactivation, visible-light bacterial inactivation has clear advantages in terms of its ease of use and substantially greater safety in a clinical setting. Thus, aPDT has arisen as an alternative therapy to antibiotics in the treatment of bacterial infections. The antimicrobial effect of aPDT is based on the principle that visible light activates an externally applied photosensitizer. This results in the generation of reactive oxygen species (ROS) that kill bacteria unselectively. The use of ICG as a photosensitizer has many benefits, including the low toxicity and the ability to release the absorbed energy as heat through internal conversion, in addition to the fluorescence emission and the ROS producing triplet state. Thus, the depth of penetration might be increased when compared to other photosensitizers. [25]. APDT with ICG has previously shown a significant antibacterial action against *S. aureus*, studied in the methicillin-resistant strains [26]. ABL is based on the same principle as aPDT, but the photosensitizers in the process are inherent molecules within the bacteria itself, such as porphyrins and flavins [18]. Of the different blue light spectrums, 405 nm aBL has been shown to outperform longer aBL wavelengths in several studies [27,28]. Light coherence has no significant role in the process, and the use of LED light sources would not make a significant difference to laser light sources [29]. 

A combination of different wavelengths to improve the bactericidal effect against *S. aureus* has been proposed earlier. In a study by Leanse et al. with a dual-wavelength aBL irradiation approach, the combination of 460 and 405 nm light was more effective against methicillin-resistant *S. aureus* than separately given irradiations. In this study, the 460 nm light was proposed to inactivate an oxygen quencher staphyloxanthin from the cell surface, which showed a protective action against 405 nm aBL exposure. [30]. Another dual-light approach was conducted by Guffrey et al., wherein the effect of a combination of 405 nm blue light and 880 nm infrared light on *S. aureus* and *Pseudomonas aeruginosa* was tested. The bacteria were treated simultaneously with a combination of 405 nm and 880-nm light. The results revealed a significant dose-dependent bactericidal effect, although the *S. aureus* experiments resulted in statistically significant decreases in bacterial colonies at all dose levels. However, the near-infrared light was applied without a photosensitizer and had only a small effect. [31]. The aforementioned findings were in line with our previous results when we compared the pure NIR light to the combined effect of NIR together with ICG [19]. In the present study, we did not test the antibacterial effect of the sole 810 nm light. 

We tested the repeated dosing of aBL and aPDT combination for two reasons. Firstly, to simulate a clinical antibacterial treatment protocol of daily dosing, and secondly, to test the ability of *S. aureus* to adapt against a repeated dual-light therapy in a short treatment setting. We have recently published data regarding the ability of *S. mutans* to adapt against both aBL and aPDT, when given separately in a repeated fashion [19]. The adaptation of *S. mutans* was not seen when aBL and aPDT were combined as a dual-light treatment. In the present study, similarly to our previous findings with *S. mutans*, we saw the retained antibacterial action abate with *S. aureus* when aBL or aPDT were used separately. The ability of *S. aureus* to adapt against the repeated application of 405 nm has also been shown by other groups [30,31]. Although in the recent study, the aPDT showed markedly better antibacterial effect compared to aBL against *S. aureus* biofilm, the biofilm showed adaptation to repeated exposure. This was seen by increased CFU counts of up to 100-fold when compared to the single-dose aPDT treatment. The response to the repeated adverse environmental stimuli seemed to develop in the early stage, within the first few repeated exposures. In our protocol, the bacteria had one day (approximately 24 h) to build responsive actions. Again, the bacteria were capable of adapting when repeated single wavelength antibacterial light treatment was applied. The dual-light aPDT against *S. aureus* markedly outperformed both aPDT and aBL in efficacy when compared to the single-use protocol, but most importantly, the synchronized use was able to suppress the ability of the biofilm to adapt to the repeated treatment protocol. This suppression gives great promise to the repeated dual-light treatment protocol in the clinical setting. 

Skin and soft-tissue infections are common problems encountered in clinical practice. An increase in the incidence of skin and soft-tissue infections has been observed during previous decades. [32]. Treatment has been significantly complicated by the increasing emergence of multidrug-resistant pathogenic bacteria, especially methicillin-resistant *S. aureus* [2]. These infections are responsible for a significant percentage of post-surgery infective complications [33,34], patient mortality, and massive healthcare costs in hospitals worldwide [35,36]. Empiric treatment considerations are likely to be changed by the increasing prevalence of antibiotic-resistant bacteria, such as methicillin-resistant *S. aureus*. Hence, future treatments for local tissue infections could include light as adjunctive therapy in the antibacterial treatment. Our results indicate that the use of a combination of aBL and aPDT indeed provides an increase in the efficacy of bacterial killing, which can be attractive in clinical practice. The combination here presented improves the effect in both single-dose treatments and in repeated protocols. An example where an efficient single treatment would be lucrative is a local antibacterial treatment during surgery, where the local access to the infection site would be limited. Such local antibacterial treatment could be, for example, the disinfection of valve annulus during endocarditis valve replacement surgery or, similarly, during endoprosthesis replacement in orthopedics. On the other hand, the improved and retained antibacterial effect at easily reachable infection sites, such as the local impetigo of superficial wound infection, would be greatly appreciated. This finding already may greatly help to extend the indicated usage of dual-light aPDT. Importantly, it paves the way forward for evaluating dual-light-based aPDT treatments for difficult-to-treat chronic wound infections or burn-wound infections complicated by polymicrobial bacterial biofilms resistant to many treatments now in use.

### Limitations

It could be argued that utilizing almost an identical study protocol to investigate the effects of aBL, aPDT, and dual-light aPDT on the viability of *S. aureus* in the present study, as performed in our previous article reporting on *S. mutans* [18], would be repetitive. On the contrary, we speculate that the use of similar light irradiances, bacterial culture, and colony-forming-unit counting methods produces valuable comparability. Eventually, in this report, we show the need to increase dual-light dosing against *S. aureus*, compared to *S. mutans*. The ability to accurately assess the requirements of successful bacterial eradication is vital during the development of light-based treatments of infections at the time of growing antibiotic resistance.

## 5. Conclusions

The objective of this proposal was to investigate the utility of a non-antibiotic approach against *S. aureus*. Our results indicate the effectiveness of a dual-light combination of aBL and aPDT in improving the antibacterial efficacy when compared with aPDT or aBL treatment alone. The dual-light efficacy was sustained in a three-day repetitive use model, and we showed the importance of the amount of aBL for optimal antibacterial efficacy. This study reveals a new effective local antibacterial method against *S. aureus*, one of the most important pathogens responsible for a significant percentage of common topical skin infections, but also post-surgery infective complications, patient mortality, and massive healthcare costs in hospitals worldwide.

## Figures and Tables

**Figure 1 antibiotics-10-01240-f001:**
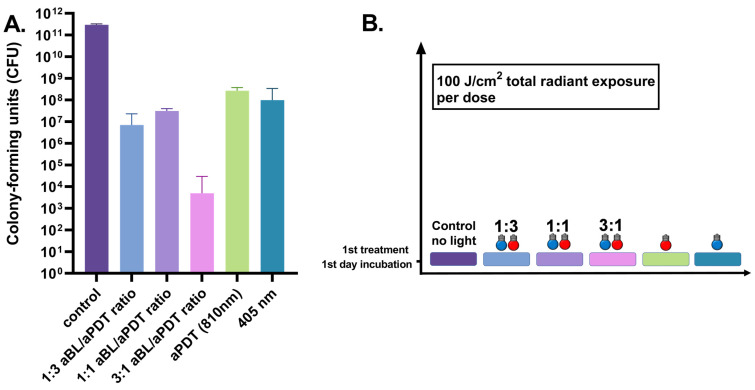
(**A**). Effects of different irradiance ratios of aBL and aPDT in the dual-light aPDT system in one-day *S. aureus* biofilm. One-day *S. aureus* biofilm, single dose, aBL vs. dual-light 1:3 aBL/aPDT ratio, *p* = 0.0022; aBL vs. dual-light 1:1 aBL/aPDT ratio, *p* = 0.015; aBL vs. dual-light 3:1 aBL/aPDT ratio, *p* = 0.0022; aPDT vs. dual-light 1:3 aBL/aPDT ratio, *p* = 0.0022; aPDT vs. dual-light 1:1 aBL/aPDT ratio, *p* = 0.015; aPDT vs. dual-light 3:1 aBL/aPDTratio, *p* = 0.0022. aBL vs. aPDT, *p* = 0.046; dual-light aPDT: 1:3 aBL/aPDT ratio vs. 1:1 aBL/aPDT ratio, *p* = ns; 1:3 aBL/aPDT ratio vs. 3:1 aBL/aPDT ratio, *p* = ns; 1:1 aBL/aPDT ratio vs. 3:1 aBL/aPDT ratio; *p* = ns. The *y*-axis displays the median of CFUs. Six assays were performed for each experiment, and the control biofilm was performed in three assays. The 95% confidence interval (CI) is presented by using t-bars. (**B**). A schematic representation of the experiment.

**Figure 2 antibiotics-10-01240-f002:**
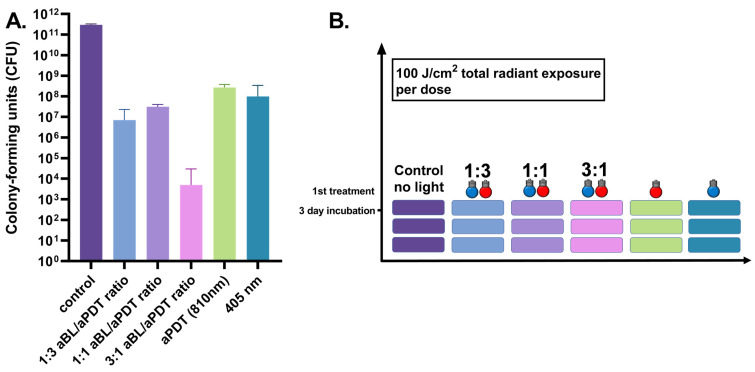
(**A**). Effects of different irradiance ratios of aBL and aPDT in the dual-light aPDT system in three-day *S. aureus* biofilm. Three-day *S. aureus* biofilm, single dose, aBL vs. dual-light 1:3 aBL/aPDT ratio, *p* = 0.17; aBL vs. dual-light 1:1 aBL/aPDT ratio, *p* = 1.0; aBL vs. dual-light 3:1 aBL/aPDT ratio, *p* = 0.0001; aPDT vs. dual-light 1:3 aBL/aPDT ratio, *p* = 0.0022; aPDT vs. dual-light 1:1 aBL/aPDT ratio, *p* = 0.0022; aPDT vs. dual-light 3:1 aBL/aPDT ratio, *p* = 0.0001; aBL vs. aPDT, *p* = 0.13; dual-light aPDT: 1:3 aBL/aPDT ratio vs. 1:1 aBL/aPDT ratio, *p* = 0.011; 1:3 aBL/aPDT ratio vs. 3:1 aBL/aPDT ratio, *p* = 0.0001; 1:1 aBL/aPDT ratio vs. 3:1 aBL/aPDT ratio; *p* = 0.0001. The *y*-axis displays the median of CFUs. Six assays were performed for each experiment, and the control biofilm was performed in three assays. The 95% confidence interval (CI) is presented by using t-bars. (**B**). A schematic representation of the experiment.

**Figure 3 antibiotics-10-01240-f003:**
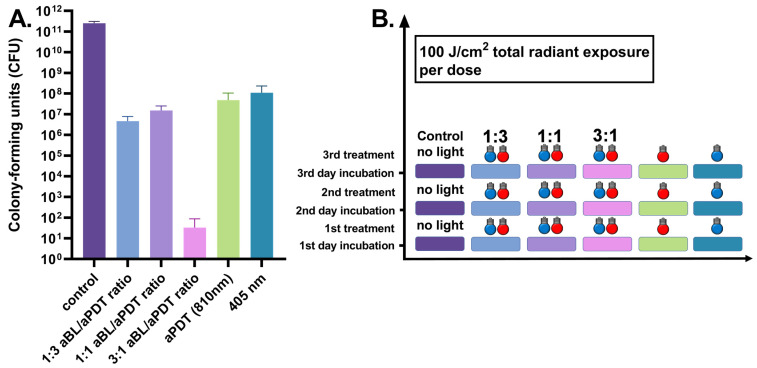
(**A**). Effects of different irradiance ratios of aBL and aPDT in the dual-light aPDT system in three-day *S. aureus* biofilm, when the antibacterial treatment is given once daily. Three-day *S. aureus* biofilm, daily dose, aBL vs. dual-light 1:3 aBL/aPDT ratio, *p* = 0.0022; aBL vs. dual-light 1:1 aBL/aPDT ratio, *p* = 0.015; aBL vs. dual-light 3:1 aBL/aPDT ratio, *p* = 0.0022; aPDT vs. dual-light 1:3 aBL/aPDT ratio, *p* = 0.0043; aPDT vs. dual-light 1:1 aBL/aPDT ratio, *p* = 0.015; aPDT vs. dual-light 3:1 aBL/aPDT ratio, *p* = 0.0022; aBL vs. aPDT, *p* = 0.26; dual-light aPDT: 1:3 aBL/aPDT ratio vs. 1:1 aBL/aPDT ratio, *p* = 0.03; 1:3 aBL/aPDT ratio vs. 3:1 aBL/aPDT ratio, *p* = 0.0022; 1:1 aBL/aPDT ratio vs. 3:1 aBL/aPDT ratio; *p* = 0.0022. The *y*-axis displays the median of CFUs. Six assays were performed for each experiment, and the control biofilm was performed in three assays. The 95% confidence interval (CI) is presented by using t-bars. (**B**). A schematic representation of the experiment.

**Figure 4 antibiotics-10-01240-f004:**
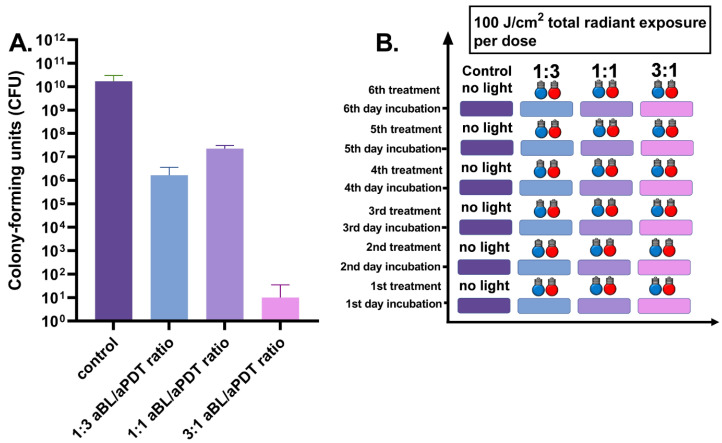
(**A**). Effects of different irradiance ratios of aBL and aPDT in the dual-light aPDT system in six-day *S. aureus* biofilm, when the antibacterial treatment is given once daily. Six-day *S. aureus* biofilm, daily-dose dual-light aPDT: 1:3 aBL/aPDT ratio vs. 1:1 aBL/aPDT ratio, *p* = 0.0022; 1:3 aBL/aPDT ratio vs. 3:1 aBL/aPDT ratio, *p* = 0.0022; 1:1 aBL/aPDT ratio vs. 3:1 aBL/aPDT ratio; *p* = 0.0022. The *y*-axis displays the median of CFUs. Six assays were performed for each experiment, and the control biofilm was performed in three assays. The 95% confidence interval (CI) is presented by using t-bars. (**B**). A schematic representation of the experiment.

**Figure 5 antibiotics-10-01240-f005:**
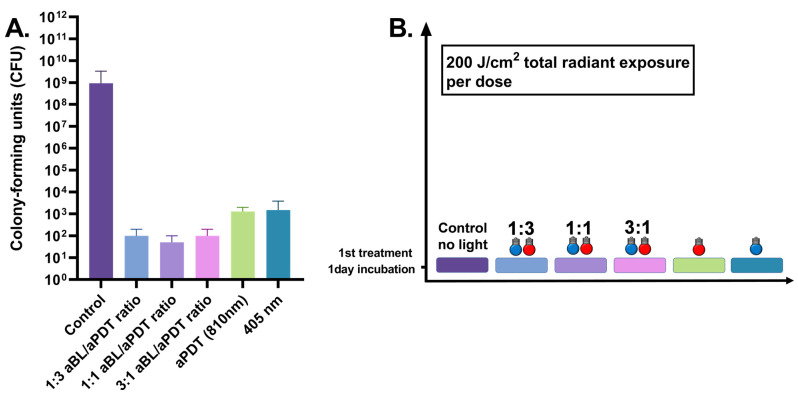
(**A**). Effects of different irradiance ratios of aBL and aPDT in the dual-light aPDT system in one-day *S. aureus* biofilm daily at 200 J/cm2. One-day *S. aureus* biofilm, single dose, aBL vs. dual-light 1:3 aBL/aPDT ratio at 200 J/cm2, *p* = 0.0022; aBL vs. dual-light 1:1 aBL/aPDT ratio, *p* = 0.0022; aBL vs. dual-light 3:1 aBL/aPDT ratio, *p* = 0.0022; aPDT vs. dual-light 1:3 aBL/aPDT ratio, *p* = 0.0022; aPDT vs. dual-light 1:1 aBL/aPDT ratio, *p* = 0.0022; aPDT vs. dual-light 3:1 aBL/aPDT ratio, *p* = 0.0022. aBL vs. aPDT, *p* = ns; dual-light aPDT: 1:3 aBL/aPDT ratio vs. 1:1 aBL/aPDT ratio, *p* = ns; 1:3 aBL/aPDT ratio vs. 3:1 aBL/aPDT ratio, *p* = ns; 1:1 aBL/aPDT ratio vs. 3:1 aBL/aPDT ratio; *p* = ns. The *y*-axis displays the median of CFUs. Six assays were performed for each experiment, and the control biofilm was performed in three assays. The 95% confidence interval (CI) is presented by using t-bars; ns = non-significant. (**B**). A schematic representation of the experiment.

**Figure 6 antibiotics-10-01240-f006:**
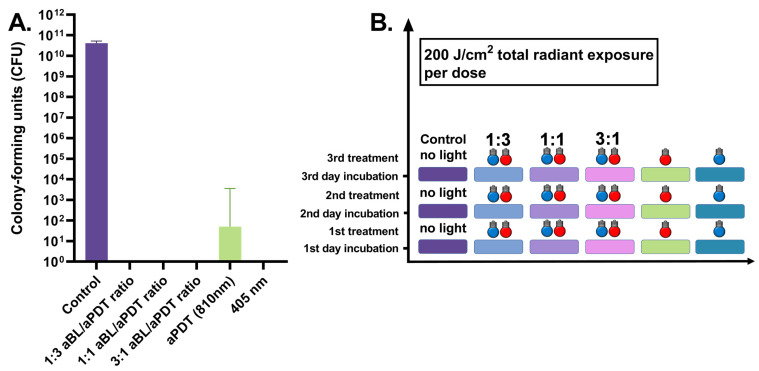
(**A**). Effects of different irradiance ratios of aBL and aPDT in the dual-light aPDT system in three-day *S. aureus* biofilm, when the antibacterial treatment is given once daily at 200 J/cm2. Three-day *S. aureus* biofilm, daily dose, aBL vs. dual-light 1:3 aBL/aPDT ratio, *p* = ns; aBL vs. dual-light 1:1 aBL/aPDT ratio, *p* = ns; aBL vs. dual-light 3:1 aBL/aPDT ratio, *p* = ns; aPDT vs. dual-light 1:3 aBL/aPDT ratio, *p* = ns; aPDT vs. dual-light 1:1 aBL/aPDT ratio, *p* = ns; aPDT vs. dual-light 3:1 aBL/aPDT ratio, *p* = ns; aBL vs. aPDT, *p* = ns; dual-light aPDT: 1:3 aBL/aPDT ratio vs. 1:1 aBL/aPDT ratio, *p* = ns; 1:3 aBL/aPDT ratio vs. 3:1 aBL/aPDT ratio, *p* = ns; 1:1 aBL/aPDT ratio vs. 3:1 aBL/aPDT ratio; *p* = ns. The *y*-axis displays the median of CFUs. Six assays were performed for each experiment, and the control biofilm was performed in three assays. The 95% confidence interval (CI) is presented by using t-bars; ns = non-significant. (**B**). A schematic representation of the experiment.

**Table 1 antibiotics-10-01240-t001:** Exact details of the test methodology.

Experiment	Figure	Repeats	Number of Treaments	Radiant Exposure (J/cm^2^)	Wavelenghts (nm)	Irradiance of 405 nm (mW/cm^2^)	Irradiance of 810 nm (mW/cm^2^)	ICG (+/−)	Biofilm Age at the End of Experiment (Days)
1-day aBL	1	6	1	100	405	80	0	−	1
1-day aPDT	1	6	1	100	810	0	100	+	1
1-day single dose 1:3	1	6	1	100	405 + 810	42	135	+	1
1-day single dose 1:1	1	6	1	100	405 + 810	73	79	+	1
1-day single dose 3:1	1	6	1	100	405 + 810	130	38	+	1
1-day control	1	3	N/A	N/A	N/A	N/A	N/A	−	1
1-day aBL	5	6	1	200	405	80	0	−	1
1-day aPDT	5	6	1	200	810	0	100	+	1
1-day single dose 1:3	5	6	1	200	405 + 810	42	135	+	1
1-day single dose 1:1	5	6	1	200	405 + 810	73	79	+	1
1-day single dose 3:1	5	6	1	200	405 + 810	130	38	+	1
1-day control	5	6	N/A	N/A	N/A	N/A	N/A	−	1
3-day aBL	2	6	1	100	405	80	0	−	3
3-day aPDT	2	6	1	100	810	0	100	+	3
3-day single dose 1:3	2	6	1	100	405 + 810	42	135	+	3
3-day single dose 1:1	2	6	1	100	405 + 810	73	79	+	3
3-day single dose 3:1	2	6	1	100	405 + 810	130	38	+	3
3-day control	2	3	N/A	N/A	N/A	N/A	N/A	−	3
3-day daily dose aBL	3	6	3	100	405	80	0	−	3
3-day daily dose aPDT	3	6	3	100	810	0	100	+	3
3-day daily dose 1:3	3	6	3	100	405 + 810	42	135	+	3
3-day daily dose 1:1	3	6	3	100	405 + 810	73	79	+	3
3-day daily dose 3:1	3	6	3	100	405 + 810	130	38	+	3
3-day control	3	3	N/A	N/A	N/A	N/A	N/A	−	3
3-day daily dose aBL	6	6	3	200	405	80	0	−	3
3-day daily dose aPDT	6	6	3	200	810	0	100	+	3
3-day daily dose 1:3	6	6	3	200	405 + 810	42	135	+	3
3-day daily dose 1:1	6	6	3	200	405 + 810	73	79	+	3
3-day daily dose 3:1	6	6	3	200	405 + 810	130	38	+	3
3-day control	6	6	N/A	N/A	N/A	N/A	N/A	−	3
6-day aBL	4	6	6	100	405	80	0	−	6
6-day aPDT	4	6	6	100	810	0	100	+	6
6-day single dose 1:3	4	6	6	100	405 + 810	42	135	+	6
6-day single dose 1:1	4	6	6	100	405 + 810	73	79	+	6
6-day single dose 3:1	4	6	6	100	405 + 810	130	38	+	6
6-day control	4	3	N/A	N/A	N/A	N/A	N/A	−	6

ICG, indocyanine green; N/A, not available.

## Data Availability

The data presented in this study are available on request from the corresponding author.

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
