# Peer review of "Daily Administered Dual-Light Photodynamic Therapy Provides a Sustained Antibacterial Effect on Staphylococcus aureus"

_antibiotics, 2021, doi:10.3390/antibiotics10101240_

Round 1

Reviewer 1 Report

The manuscript entitled: “Daily administered dual-light photodynamic therapy provides a sustained antibacterial effect on Staphylococcus aureus”, reference: antibiotics-1400719

General comments:

In my understanding the manuscript is unquestionably important, thus its findings should be reported. Nevertheless, in my opinion the manuscript has several critical issues that should be addressed. Namely a grievous lack of information in the Material and Methods section. Even though reported elsewhere, it was using a different bacterium. Also I failed to find proofs of biofilm formation, do you understand my perspective? In my opinion, the authors should detail all the methodology and preferably include a diagram for the manuscript to become even more reader friendly and greatly enhance its impact.

Moreover, and for me this a key question, the energy density used by the authors is in my point of view: brutal. I beg your pardon for the term, but I could not find a better one. How much time was required to obtain such dose? If it required a very long time will it still be feasible to be used as treatment? Do the authors understand my point view?

Point by point comments:

Line 39, in the Keywords section please revise the name of specie Staphylococcus aureus, since the specific epithet should not be capitalized.

Line 48, "with only some positive change from the 1940’s to 2010" I understand what the authors intend to say, however, in my opinion, the clarity of this sentence bay be considerably improved.

Line 51 to 52, in my opinion, if the authors included additional of the referred strains it would improve the manuscript impact.

Line 54 and 55, “Novel antibiotics are difficult and expensive to develop” in my opinion the cost and difficulty are not the main factors for the development of novel antibiotics. It was extremely expensive and difficult to come up with vaccines for SARS-CoV-2, nevertheless we made it. Please provide additional hypothesis and please adequately support them with adequate literature.

Line 68, “the essential means in ROS clearance” not entirely clear statement, please revise.

Line 79, please italicize all species names.

Material and Methods section is in my opinion is grievously incomplete. In my opinion it is impossible to replicate the performed experiments. Moreover, several important details are missing, namely: the bacterium specie used comes from a collection or is it an isolate, what were the incubation conditions (medium, temperature, etc), the authors mention biofilm how was it assessed? The energy density how was it determined? How were the tests actually performed? How much area was exposed to the light? What was the material where the bacterium as inoculated during the treatments? All these questions and more are left unanswered. I understand that the authors refer that “presented accurately elsewhere [15, 16]”, but in my modest opinion each article should possess the entire critical information, for the readers to be able to replicate, and not be forced to search that information in other documents. Mainly because in some documents that information may be lost.

Line 101 to 102, I am having problems on how the authors assessed and confirmed biofilm formation, in addition how the colony forming units were assessed from a biofilm. Any techniques were applied for the disruption of extracellular polymeric substances of the biofilm without compromising the cells viability? Moreover, I do not understand why the authors did not estimate the concentration of bacterium, meaning CFU/mL for instance, which would be universal unlike just CFU. I am confused, please enlighten me. Maybe all this questions could be answered in materials and methods section.

Line 107, in my opinion the authors should use the CFU in a more reader friendly way, meaning instead of “0.06x10^6-3.4x10^6” it should be 6.0x10^4 – 3.4x10^6. In my understanding the log difference is more directly perceived this way.

Figure 1, in my opinion, the figures would be easier to read if the control results were placed as the first column from the left, as the vast majority of the publications exhibit this order. Please revise all figures.

  Since all figures exhibit the same range of results, all figures Y-axis should be the same. It would also make easer to compare.

Author Response

Reviewer 1:

General comments:

In my understanding the manuscript is unquestionably important, thus its findings should be reported. Nevertheless, in my opinion the manuscript has several critical issues that should be addressed. Namely a grievous lack of information in the Material and Methods section. Even though reported elsewhere, it was using a different bacterium. Also I failed to find proofs of biofilm formation, do you understand my perspective? In my opinion, the authors should detail all the methodology and preferably include a diagram for the manuscript to become even more reader friendly and greatly enhance its impact.

Dear reviewer 1, we kindly thank you for your work. The issues you bring up are relevant and definitely improve our paper.

We have rewritten the material and method section as seen below. We used ATCC 25923 strain, which has been used as a standard in several papers regarding biofilm formation and thus our paper is based on literature on this issue.  Please see “ The present study aimed to investigate the effects of aBL (405 nm), aPDT (810 nm combined with a photosensitizer: ICG), and dual-light antibacterial photodynamic therapy (405 nm and 810 nm combined with a P.S.) on monospecies S. aureus ATCC 25923 incubated for one day before any treatment. This strain is a known biofilm-forming strain [20] The formed biofilms were divided into subgroups according to the total incubation time and the treatment method. The experiments and the number of assays are shown in Table 1.”

We have included diagrams at each figure to improve the readability of the study setups.

Comment 1:

Moreover, and for me this a key question, the energy density used by the authors is in my point of view: brutal. I beg your pardon for the term, but I could not find a better one. How much time was required to obtain such dose? If it required a very long time will it still be feasible to be used as treatment? Do the authors understand my point view?

Answer: We understand the reviewer’s point of view. Examples of the irradiance times were included in the Materials and Methods chapter. Moreover, Table 1 contains all information required for the reader to be able to calculate the irradiance times for all biofilm tests.

In our opinion the irradiance time is acceptable especially when comparing it to troubles caused by multiresistent bacteria. Furthermore, the light sources could be multiplied to obtain faster treatment times.

“The irradiances of used light, the wave lengths, the number of treatments and incubation repeats in each biofilm study are presented in Table 1. For example, irradiance times of 1250 seconds and 1000 seconds were required to reach the desired radiant exposure of 100J/cm2 for the aBL and aPDT respectively.”

Point by point comments:

Comment 2:

Line 39, in the Keywords section please revise the name of specie Staphylococcus aureus, since the specific epithet should not be capitalized.

Answer: The keyword line was changed to Staphylococcus aureus as suggested.

Comment 3:

Line 48, "with only some positive change from the 1940’s to 2010" I understand what the authors intend to say, however, in my opinion, the clarity of this sentence bay be considerably improved.

Answer: We agree with the reviewer. The sentence now reads as: “The morbidity and mortality from S. aureus bacteremia have only minor improvement when comparing the reports from the 1940s to 2010s ”

Comment 4:

Line 51 to 52, in my opinion, if the authors included additional of the referred strains it would improve the manuscript impact.

Answer: We added an example of a common MRSA strain and its reference as suggested by the reviewer.

Comment 5:

Line 54 and 55, “Novel antibiotics are difficult and expensive to develop” in my opinion the cost and difficulty are not the main factors for the development of novel antibiotics. It was extremely expensive and difficult to come up with vaccines for SARS-CoV-2, nevertheless we made it. Please provide additional hypothesis and please adequately support them with adequate literature.

Answer: We revised the text accordingly: “Inappropriate antibiotic use is one of the primary reasons for antimicrobial resistance formation [110], and multiple challenges relate to the novel antibiotic drug discovery and development. A recent cost estimate of new antibiotics development is at 1.5 billion U.S. dollars on average [12].  Moreover, the timeline from its discovery to launching a commercial drug can be as long as 10 to 12 years. Finally, bacteria are capable of developing resistance to new antibiotics relatively quickly [13]. Thus, investments for research and development of new antibiotics are diminishing [14].  The need for new solutions has never been more urgent.”

Comment 6:

Line 68, “the essential means in ROS clearance” not entirely clear statement, please revise.

Answer: The sentence was changed to: “The inherent lack of catalase enzyme in streptococci, which is the bacteria’s main factor in clearing hydrogen peroxide-type reactive oxygen species from the cell, makes streptococcus very vulnerable to aPDT”

Comment 7:

Line 79, please italicize all species names.

Answer: The species names were italicized.

Comment 8:

Material and Methods section is in my opinion is grievously incomplete. In my opinion it is impossible to replicate the performed experiments. Moreover, several important details are missing, namely: the bacterium specie used comes from a collection or is it an isolate, what were the incubation conditions (medium, temperature, etc), the authors mention biofilm how was it assessed? The energy density how was it determined? How were the tests actually performed? How much area was exposed to the light? What was the material where the bacterium as inoculated during the treatments? All these questions and more are left unanswered. I understand that the authors refer that “presented accurately elsewhere [15, 16]”, but in my modest opinion each article should possess the entire critical information, for the readers to be able to replicate, and not be forced to search that information in other documents. Mainly because in some documents that information may be lost.

Answer:  We agree with the reviewer. We included a detailed description of the study methodology in the Materials and Methods chapter. “The methodology in the present study is presented in the following steps. Step 1: The Staphylococcus aureus strain (ATCC  25923) growth in a BHI broth (Bio-Rad 3564014, Bio-Rad Laboratories iInc, US.S.). Step 2: Incubation at +36 degrees C in an air concentra-tion of 5% CO2 for 18 hours (NuAire DH autoflow 5500, NuAire iInc, US.S.). Step 3: Dilu-tion to the optical density of 0.46 using 0.9% NaCl solution (measured using a spectropho-tometer, Varian Cary 100 Bio UV-VIS, Agilent Technologies Iinc, US.S. and Den 1 McFar-land Densitometer, Biosan, Riga, Latvia). Step 4: Biofilm growth in a flat-bottom 96-well plate (Thermo Fisher Scientific Iinc, US.S.). Step 5: Placing 100 l of S. aureus suspension into the well-plates, each primed with 100 l of BHI broth. (Additional steps 5.1: incuba-tion at a temperature of 36 degrees C and in an air concentration of 5% CO2 in addition to a daily change of the 100 l BHI broth to a fresh solution). Step 6: Removal of Ggrowth medium  by pipetting and replacement with indocyanine green solution 250 g/ml (Verdye, Diagnostic Green, GmBH). Step 7: Incubation in a dark room for 10 min. Step 8: Washing the biofilm with 0.9% NaCl solution until each well contained a total volume of 200 l. Step 9: Light exposure using a custom-made LED light (Lumichip Oy, Espoo, Fin-land), the time of exposure was calculated using the known irradiances of the light sources (measured using a light energy meter, Thorlabs PM 100D and an S121C sensor head, Thorlabs Iinc, US.S., in addition to a spectroradiometer, BTS256, Gigahertz-Optik GmBH, Germany). The irradiances of used light, the wave lengths, the number of treat-ments, and incubation repeats in each biofilm study are presented in Table 1. For example, irradiance times of 1250 seconds and 1000 seconds were required to reach the desired ra-diant exposure of 100J/cm2 for the aBL and aPDT, respectively. Step 10: Changing of the BHI broth and subsequent incubation or biofilm removal for colony forming unit counting if the planned study exposure was finished. Step 11: After the light exposure, removal of the entire biofilms into a 1-ml test tube, forming 200 l of solution. Step 12: Vortexing us-ing a (Vortex-Genie, Scientific Industries Iinc, US.S.) and serial dilution ranging from 1:1 to 1:100 000 using a sterile tipped ART filters (Thermo Scientific, Waltham, US.S.). Step 13: Spreading of 100 l of the resulting biofilm dilution onto a BHI agar- plate using a sterile L-rod. The dilutions were performed according to the observed biofilm mass from the well-plates. Dilution where the CFU counts were between 30 and 800, where the CFU counts were between 30 and 800, were selected for analysis, whereas the results for CFU 0 analysis were obtained from a solution with a 1:1 dilution factor. Step 14: Incubation for 48 hours and photographing of the plates (Leica TCS CARS SP 8X microscope, Leica Mi-crosystems, Wetzlar, Germany) with HC PL APO CS2 20X/0.75 numerical-aperture multi immersion and HX PL APO CS2 63X/1.2 numerical-aperture water immersion objectives. Step 15: Staining with a livae/dead BacLight bacterial viability kit (Molecular Probes, Invi-trogen, Eugene, Oregon, US.S.) and incubation in a dark room for 15min. Step 16: Exami-nation under a confocal scanning laser microscope, with a two-laser system (488nm ar-gon laser and a 561nm DPSS laser). The emission windows for the 488nm and 561nm la-sers were set at 500nm-530nm and 620nm-640nm, respectively. The institutional review board of Helsinki University Hospital (HUS) provided an ethical approval for the present study.  ”

Comment 9:

Line 101 to 102, I am having problems on how the authors assessed and confirmed biofilm formation, in addition how the colony forming units were assessed from a biofilm. Any techniques were applied for the disruption of extracellular polymeric substances of the biofilm without compromising the cells viability? Moreover, I do not understand why the authors did not estimate the concentration of bacterium, meaning CFU/mL for instance, which would be universal unlike just CFU. I am confused, please enlighten me. Maybe all this questions could be answered in materials and methods section.

Answer: We agree with the reviewer’s comments. The questions are answered in the revised Materials and Methods chapter.

Comment 10:

Line 107, in my opinion the authors should use the CFU in a more reader friendly way, meaning instead of “0.06x10^6-3.4x10^6” it should be 6.0x10^4 – 3.4x10^6. In my understanding the log difference is more directly perceived this way.

Answer: The number of CFUs are now presented as suggested by the reviewer.

Comment 11:

Figure 1, in my opinion, the figures would be easier to read if the control results were placed as the first column from the left, as the vast majority of the publications exhibit this order. Please revise all figures.

Answer: The control results were changed to the left side.

Comment 12:

  Since all figures exhibit the same range of results, all figures Y-axis should be the same. It would also make easer to compare.

Answer: All figures were corrected according to the reviewer’s suggestions.

Reviewer 2 Report

Dear Authors,

The present study evaluates the use of dual-light Antimicrobial Photodynamic therapy against S. aureus. The research subject is interesting and brings scientific important data in the field, as it deals with an important matter nowadays related to the fight against antimicrobial resistance. Some changes of the manuscript should nevertheless be performed in order to improve its quality. Following specific changes should thus be performed:

Minor changes

            Please remove name of sections from the abstract.

            All names of genera and bacteria should be italic. Please check.

Major changes

Abstract is too long. Please reduce to 200 words.

The structure that is needed for the manuscript is not followed. Please use the template on the journal site and follow the structure you find there.

Introduction should contain data on similar studies existing in scientific literature and, in comparison, authors should emphasize the novelty and originality of their study. These informations need to be added in this part. Make connections between your different parts of the Introduction section.  

 Please add informations on the Materials and Methods you use, there is no relevant information on the related section. This section needs serious improvement. Even if the methods are described elsewhere, authors should find a way to present them.

All these suggested changes should be performed in order to bring further improvements to the manuscript.

Author Response

Reviewer 2:

The present study evaluates the use of dual-light Antimicrobial Photodynamic therapy against S. aureus. The research subject is interesting and brings scientific important data in the field, as it deals with an important matter nowadays related to the fight against antimicrobial resistance. Some changes of the manuscript should nevertheless be performed in order to improve its quality. Following specific changes should thus be performed:

Comment 1:

Minor changes

            Please remove name of sections from the abstract.

            All names of genera and bacteria should be italic. Please check.

Answer: All names of the sections have been removed from the abstract and the names and genera of bacteria are changed to italic.

Comment 2:

Major changes 

Abstract is too long. Please reduce to 200 words.

The structure that is needed for the manuscript is not followed. Please use the template on the journal site and follow the structure you find there.

Answer: The abstract has been shortened to 199 words as suggested.

Comment 3:

Introduction should contain data on similar studies existing in scientific literature and, in comparison, authors should emphasize the novelty and originality of their study. These informations need to be added in this part. Make connections between your different parts of the Introduction section.  

Answer: The introduction section was revised as suggested by the reviewer.

Comment 4:

 Please add informations on the Materials and Methods you use, there is no relevant information on the related section. This section needs serious improvement. Even if the methods are described elsewhere, authors should find a way to present them.

Answer: We agree with the reviewer. The Materials and Methods chapter has been re-written and it now contains step-by-step instructions of our test methodology.

All these suggested changes should be performed in order to bring further improvements to the manuscript.

Round 2

Reviewer 1 Report

Excellent work, congratulations and thank you very much.

Author Response

We thank the reviewer so far for his/her time and efforts in improving our manuscript.

Reviewer 2 Report

Dear Authors,

The present study evaluates the use of dual-light Antimicrobial Photodynamic therapy against S. aureus. Authors performed all the suggested changes after the first round of review and properly included them into the manuscript. Some changes should still be performed in order to improve its quality. Following specific changes should thus be performed:

 Minor changes

            Please remove phrases as “Please see Figure 1”. It is enough to put at the end of the phrase name of figure between brakets (Fig. 1).

            All names of genera and bacteria should be italic. Please check. Moreover, there are places where S. aureus has the second letter still in capital letter and it shouldn’t. please check (e.g. line 168)

Major changes

Introduction should still emphasize the novelty and originality of their study and make connections between your different parts of the Introduction section.  

All these suggested changes should be performed in order to bring further improvements to the manuscript. 

Author Response

Dear editor and reviewers.

We thank the reviewers so far for their time and efforts in improving our manuscript.

Dear Authors,

The present study evaluates the use of dual-light Antimicrobial Photodynamic therapy against S. aureus. Authors performed all the suggested changes after the first round of review and properly included them into the manuscript. Some changes should still be performed in order to improve its quality. Following specific changes should thus be performed:

 Minor changes

            Please remove phrases as “Please see Figure 1”. It is enough to put at the end of the phrase name of figure between brakets (Fig. 1).

We have removed the phrase and corrected as suggested. Line 157

            All names of genera and bacteria should be italic. Please check. Moreover, there are places where S. aureus has the second letter still in capital letter and it shouldn’t. please check (e.g. line 168)

We have rechecked this in the text as well in the references.

Major changes

Introduction should still emphasize the novelty and originality of their study and make connections between your different parts of the Introduction section.

We have connected the idea of the first chapter to the second chapter and reduced the second chapter to keep the focus. We emphasize the  originality of our work already at the end of the third chapter, but we re-emphasize the originality at the revised fourth chapter too.

All these suggested changes should be performed in order to bring further improvements to the manuscript.